# Work-related stress: the impact of COVID-19 on critical care and redeployed nurses: a mixed-methods study

Janice Rattray [iD],[1] Louise McCallum,[2] Alastair Hull,[3] Pam Ramsay,[4] Lisa Salisbury,[5] Teresa Scott,[6] Stephen Cole,[7] Jordan Miller,[1] Diane Dixon[1]

► Prepublication history and additional online supplemental material for this paper are available online. To view these files, please visit the journal online. To view these files, please visit the journal online (http://dx.doi.org/10.1136/bmjopen-2021-051326).

## ABSTRACT

**Introduction** We need to understand the impact of COVID-19 on critical care nurses (CCNs) and redeployed nurses and National Health Service (NHS) organisations.
**Methods and analysis** This is a mixed-methods study (QUANT-QUAL), underpinned by a theoretical model of occupational stress, the Job Demand-Resources Model (JD-R). Participants are critical care and redeployed nurses from Scottish and three large English units.
Phase 1 is a cross-sectional survey in part replicating a pre-COVID-19 study and results will be compared with this data. Linear and logistic regression analysis will examine the relationship between antecedent, demographic and professional variables on health impairment (burnout syndrome, mental health, post-traumatic stress symptoms), motivation (work engagement, commitment) and organisational outcomes (intention to remain in critical care nursing and quality of care). We will also assess the usefulness of a range of resources provided by the NHS and professional organisations.
To allow in-depth exploration of individual experiences, phase 2 will be one-to-one semistructured interviews with 25 CCNs and 10 redeployed nurses. The JD-R model will provide the initial coding framework to which the interview data will be mapped. The remaining content will be analysed inductively to identify and chart content that is not captured by the model. In this way, the adequacy of the JD-R model is examined robustly and its expression in this context will be detailed.
**Ethics and dissemination** Ethics approval was granted from the University of Aberdeen CERB2020101993. We plan to disseminate findings at stakeholder events, publish in peer-reviewed journals and at present at national and international conferences.

## Strengths and limitations of this study

► The study has a robust theoretical framework.
► We can compare our survey results with pre-COVID-19 data.
► Our qualitative phase will provide an in-depth account of critical care nurse and redeployed nurses' experiences of working during the pandemic.
► We will be able to provide National Health Service managers with information that will provide a basis for supporting nurses during subsequent waves and future pandemics.
► Our findings will be limited to nurses only, thus excluding other healthcare workers

For numbered affiliations see end of article.

**Correspondence to**
Dr Janice Rattray;
janice.rattray@abdn.ac.uk

## INTRODUCTION

The contribution made by critical care nurses (CCNs) and those redeployed to critical care areas during this pandemic has been vital, and their expertise in caring for the increased number of critically ill patients, essential; thus, their well-being to allow them to continue in these roles is crucial. It has become clear during the first and now subsequent waves of COVID-19, that CCNs have been working in a highly charged environment with additional challenges such as: supervising redeployed staff with limited or no critical care experience; the high mortality rate of COVID-19 patients; delivering care using personal protective equipment; communicating with and supporting relatives at a distance, and the well-publicised potential risks to personal and family health. This may lead to an increase in work-related stress and its consequences at individual, unit and organisational levels. Staff redeployed to intensive care units will face these issues also but have the added challenges of an unfamiliar environment and may feel they do not have the required skill set to care for these severely ill patients. In general, work-related stress in CCNs can lead to a range of physical[1] and psychological[2] sequalae that may present as 'burnout'. It has been reported that the prevalence of burnout in CCNs prior to COVID-19 was around 16%–33%, resulting in negative outcomes such as reduced quality of care, increase in staff sickness and increased staff turnover.[3–5] Previously identified factors associated with CCN work-related stress are

based on a poor-quality literature; tend to be inconsistent, with poor conceptualisation of work-related stress and a lack of an underpinning theoretical framework. Notwithstanding those concerns, it has been suggested that factors can be described as individual, job and work environment characteristics.

It is not unreasonable to assume that the COVID-19 pandemic and its intensified challenges will result in increased CCN work-related stress. In a recent study 45% of critical care staff met the threshold for at least one of severe depression and/or anxiety, Post-traumatic stress disorder (PTSD) or problem drinking, and nursing staff were more likely to report higher levels of distress than other staff.[6] However, Greenberg's study was unable to compare current levels of distress to those prior to the pandemic. To appreciate fully the impact of this pandemic, we need to understand how work-related stress has changed and evolved from pre-COVID-19 to post-COVID-19. It is essential to do this when staff are still able to recall experiences with some clarity but, informed by the understanding that individuals, including healthcare responders, experience a broad range of early reactions, including transient distress. Consistent with the principles of Psychological First Aid we wished to allow the normalisation of these immediate responses and fostering of adaptive functioning, and instead we are interested in measuring the enduring consequences of working through the pandemic.[7] Having pre-COVID-19 data provides a comparator that can help us understand these consequences for both the individual and organisation. By using a similar cross-sectional approach and many of the same measures including burn-out our pre-COVID-19 data will allow us to do this. Further, National Health Service (NHS) boards and professional organisations have provided a range of resources to support staff (eg, self-help guides, intensive care society well-being resources) all well-meaning, but crucially with little or no evaluation. If we are to support CCNs effectively, we need urgently to understand the impact of COVID-19, evidence the specific stressors and their importance, and how these interact and subsequently impact on CCNs and their organisations. Without this knowledge, we cannot identify confidently an appropriate range of measures to protect this vital workforce, or indeed for whom, when and how to implement them.

Our study has three central strengths. First it is theoretically informed by the Job Demand-Resource Model (JD-R).[8] The JD-R model allows us to measure, understand and test a range of individual factors (personal resources, eg, resilience), work environment and job characteristics (job demands (eg, workload) and job resource variables (eg, autonomy) that may lead to either negative (health impairment, reduced job satisfaction, burn-out, impaired mental well-being) or positive (work engagement, commitment) outcomes for staff, and importantly organisational outcomes (turnover, patient safety culture and quality of care). Second, we (LM) have baseline data from a national study of work-related stress in CCNs conducted prior to the pandemic that used the JD-R model. This prepandemic data will form a baseline comparator dataset for the current study. Third, qualitative interviews will enable CCNs and those nurses redeployed to critical care to express in their own words how working during the pandemic has impacted on them. In addition, we will evaluate the support services offered to staff over the course of the pandemic to establish whether they met staff needs when delivered and addressed the sources of stress identified by the JD-R model.

The aim of this study is to establish the: (1) impact of COVID-19 on CCNs, and those nurses redeployed to critical care units and (2) prevalence of work-related stress and the perceived impact on quality of care and intention to remain in nursing. Secondary aims are to explore in detail the experiences of CCNs' and those nurses redeployed to critical care units, during the COVID-19 pandemic and to understand which service initiatives were accessed and their perceived usefulness.

Ethics approval was granted from the University of Aberdeen CERB2020101993.

## METHODS

This is a two-phase mixed-methods study (QUANT-QUAL). Phase 1 is a cross-sectional survey recruiting participants from across Scottish intensive care units, and three large English units. This phase will replicate our prepandemic study and use a range of validated and theoretically informed measures. Phase 2 will be in-depth one-to-one interviews.

Participant involvement: We have involved CCNs at different stages of this study. A current CCN (TS) is a coapplicant and has been involved in the design and implementation of the study; prior to finalising survey content we will 'sense check' this with several CCNs; we have a named CCN contact in each participating critical care unit; and four CCNs are members of the study steering group. The study commenced on 1 October 2020 with a planned duration of 12 months.

### Participants, both phases
#### Critical care nurses
CCNs employed within intensive care units (ICUs) caring for patients with level three care requirements across adult critical care units in NHS Scotland and three units in England.

#### Nurses redeployed to critical care areas
Those registered nurses (RNs) who were redeployed to critical care areas on at least two occasions.

For both CCNs and redeployed nurses: Inclusion criteria are nursing and midwifery council (NMC) RNs with substantive part-time and full-time contracts. Exclusion criteria are unregistered staff with caring roles (auxiliary/support workers), RNs on permanent agency/bank contracts.

## Sample size

### Survey

Power calculations have been calculated for the two mental health outcomes (post-traumatic stress symptomatology and mental health). A sample size of 500 (achieved in the pre-COVID-19 study with fewer units involved) will provide adequate power (80%) to detect a small effect in the GHQ-12 and the estimated prevalence of PTSD (24%) with a precision of 0.035 and confidence of 95%. Scotland has a less ethnic and racial diversity when compared with other parts of the UK. Therefore, the three additional units selected to take part in the project were identified for inclusion based, not only on their high admission rates, but also on their ability to better represent the experiences of staff from a diversity of ethnic and racial groups. The total sample eligible to participate in the pre-COVID-19 study was approximately 1224. With the inclusion of additional units, an eligible sample of approximately 2500 will be available for the current study (CCNs and redeployed nurses). The previous study achieved a recruitment rate of 48%, which if achieved here would provide a sample of n=1200.

### Interviews

We will interview a purposive sample of 25 CCNs and 10 redeployed nurses. A sampling frame will ensure we recruit from a range of gender, grades, backgrounds and ethnicity.

## Measures

### Demographic, professional and work environment details

Demographic variables will include: age, gender, relationship status, number of children, caring responsibilities, RN band/grade, tenure on the unit, number of years nursing experience and critical care nursing experience, highest level of qualification, full/part time work and shift length.

### Measures of the JD-R model

We will replicate the validated measures used to operationalise the JD-R model in the pre-COVID-19 study. These will include a range of job demands, job and personal resources, health impairment, motivational and organisational outcomes. In addition, a measure of post-traumatic stress symptoms will be included, as this has emerged in recent literature.[6] The survey will be sense checked with six CCNs to ensure that we have not missed important variables.

### Job characteristics

Job characteristics will be measured mainly using subscales from the Questionnaire on the Experience and Evaluation of Work (QEEW V.2.0)[9] This measure consists of a number of subscales that measure different aspects of workload and job stress. It has been used internationally across a range of occupational groups and each subscale has demonstrated reliability and content validity.[9] Job demands are those aspects of a job which require physical and/or mental effort and may exert physical,

psychological and cognitive effects on an employee.[10] Job resources are aspects of a job that influence goal achievement, reduce job demands and encourage personal growth and development.[10] Personal resources are characteristics that influence how well an individual deals with their work environment and how well they can control this environment.[11]

► Job Demands include seven QEEW V.2.0[9] subscales (32 items). These include: Pace and Amount of Work is a 6-item subscale relating to the speed and pressure of work alongside the amount. Emotional Load contains five items relating to how emotionally demanding work is. Mental Load contains three items relating to cognitive demand and precision of work. Physical effort includes three items relating to the physical demands of work. Complexity of work includes three items relating to the complexity and difficulty of work. Work organisation contains six items related to interruptions and hindrances in conducting work. Role conflict has five items related to aspects of a role that are disliked or unclear. Items included in these subscales are rated on a 4-point frequency response format from 'always' to 'never'. Two profession specific subscales (13 items) emerged from a previous systematic review were adapted from the Customer-Related Social Stressors questionnaire.[12] These are 'disproportionate relative/visitor expectations' and contains seven items capturing unrealistic demands from relatives and verbal aggression from relatives/visitors which has six items. Both subscales have established reliability.[12]

Job Resources include twelve QEEW V.2.0[9] subscales (52 items) and include: Learning Opportunities which is a 3-item subscale related to opportunities for growth and development. Effectiveness in achieving goals contains 4-items related to clarity of what needs to be achieved and organisational support to meet these goals. These items are scored on a 5-point 'Likert-type' scale from 'strongly agree', to 'strongly disagree'.

Autonomy has four items related to having freedom to decide or organise activities. Task clarity is also a four-item subscale related to the demarcation of responsibility for specific tasks. Feedback contains four items that relate to opportunities to obtain feedback on the purpose and results of an individual's work. Relationship with supervisor and relationship with colleagues both have six items with the former relating to an individual's relationship with their supervisor, and the latter reflecting support and collegial nature of relations within the team. Items are rated on a 4-point frequency response format from 'always' to 'never'.

Quality is an organisational-related job resource that includes four items related to the extent that quality is valued within the organisation. The final two subscales represent employment-related job resources and include well-being focus which has five items related to the extent that the organisation prioritises and

values staff well-being. Both quality and well-being focus have a 'likert-type' 5-point response format from 'strongly agree' to 'strongly disagree'. Finally, staff which has four items related to the extent that staffing levels are prioritised within the organisation and has a 4-point frequency response format.

► Personal Resources includes resilience which is the extent to which an individual prospers in the face of hardship and reflects coping ability. The measure used to assess resilience is the 10-item Connor Davidson Resilience Scale.[13] This is a single construct scale with a 5-point response format ranging from 'not true at all' to 'true nearly all of the time'.

## Outcome measures

These reflect outcomes for individuals reflecting both the negative (health impairment), and positive (motivation) arms of the JD-R model and include also organisational outcomes.

## Health impairment

► Burn-out syndrome will be measured using the 'Maslach Burnout Inventory (MBI) for Health Services Survey'[14] (22 items), a three-factor construct, capturing emotional exhaustion (the extent to which a person feels exhausted or overwhelmed by their work), depersonalisation (captures feelings or impersonal responses towards recipients) and personal accomplishment (feelings of competence and achievement). The MBI has demonstrated reliability and construct validity.[15]

► Post-traumatic stress symptoms will be measured using the post traumatic stress disorder checklist (PCL-5).[16] This 20-item measure captures the diagnostic statistical manual of mental disorders (DSM-V) symptoms (intrusive thoughts, avoidant behaviours, negative changes in thinking and mood and changes in physical and emotional reactions), consistent with the diagnosis of PTSD.[17] Participants are asked to consider how bothersome each item has been over the past month using a 4-point response format ranging from 'not at all' to 'extremely'. The PCL-5 has demonstrated reliability and validity.[16]

► Mental health using the GHQ-12 (12 items)[18] a measure of mental health well-being. The general health questionnaire (GHQ) has one domain,[19] with a four-point response category, the wording of which differs depending on the item.

Other individual outcomes will be assessed using the QEEW V.2.0[9] including Recovery after Work a 6-item subscale that relates to the immediate effects of work on home life, and Detachment after Work a 3-item subscale relating to the ability to psychologically disconnect themselves from work, hence being able to replenish resources, aid recovery and maintain health, consistent with Hobfoll's Conservation of Resources Theory (1989).[20]

## Motivation

► Work engagement refers to a high energy, positive and fulfilling work-related state of mind and will be captured using the Utrecht Work Engagement Scale[21] (9 items). This reliable scale[22] has a three-factor structure: vigour (high energy and resilience), dedication (sense of pride and commitment) and absorption (concentration and immersion within work).

► Job satisfaction using (1-item) from the QEEW V.2.0[9]

## Organisational outcomes

Intention to remain in critical care will be assessed using two subscales from the QEEW V.2.0.[9] Turnover is assessed using two subscales. Certainty about future has three items relating to how certain an employee is that they will be in the same position within the next year. Changing jobs also has three items and reflects the intention an individual has to keep or change their current job. Both have a 5-point 'Likert-type' response format from 'strongly agree to strongly disagree'.

Patient safety culture was assessed using a single item from the Agency for Healthcare Research and Quality Hospital Survey[23] that asks respondents to rate overall quality of patient safety from 'failing' to 'excellent'. Quality of care was assessed using a 15-item questionnaire 'perceptions of care left undone[24] and a single item measure of perceived quality.[25] Perceptions of care undone, asks nurses to indicate using a binary format of 'yes', 'no', against 13 activities whether care was not performed, and the number of occasions on which this occurred. The single item measure asked about in general how was the quality of nursing care from a 4-point response format ranging from 'poor' to 'excellent'. This item has performed well in previous studies involving nurses.[25]

## Support services

Participants will also be asked to identify supportive resources provided by their Health Board/NHS Trust during the pandemic. They will be asked also, how often they used these resources, their accessibility, how useful they found them, and to identify any gaps in this provision.

## Unit level data

We will collect unit level data to explore cross unit differences. This will include number of ICU beds, staffing levels, number and pattern of ICU admissions, both COVID-19 and non-COVID-19, length of ICU stay, patient mortality rates, acute physiology and chronic health evaluation (APACHE) II scores and details of service reconfiguration.

## Phase 1: theory-based survey of occupational stress
### Recruitment

► ICU managers in each of the critical care units will be contacted to (1) identify a designated unit contact/champion (this approach worked well in the initial (pre-COVID-19) study (LMcC) and (2) determine

the number of eligible CCNs within each unit and the number of redeployed nurses.

► The research team will contact and meet with each designated unit champion (either virtually due to the pandemic or in person) to introduce the study and clarify their role/responsibilities.

► The survey questionnaire will take approximately thirty minutes to complete.

### Paper based responses

► Blank, sealed questionnaire packs including the survey, a participant information sheet, a form to complete for those interested in taking part in an interview and two return envelopes (survey and interview form) will be sent to unit champions to distribute to eligible participants.

► Unit champions will identify and arrange for the questionnaire packs to be distributed to CCNs and deployed nurses and identify a return area for completed packs.

► Posters describing the study will be provided to each champion around 2 weeks prior to data collection.

► Completed questionnaires will be placed within a sealed container on each ICU and a suitable place for redeployed staff.

► Sealed containers will be packaged by the unit champion 3 weeks after recruitment commences in the unit/hospital for collection by courier and delivery to the research team.

### Online responses

► For those wishing to complete an online version of the survey, a link will be provided on the patient information sheet (PIS) and on the paper questionnaire. In addition, a uniform resource locator (URL) link and a quick response (QR) code to the online version will be displayed on the study posters.

### Consent

Consent will be assumed by the return of completed anonymised questionnaires. Unit identifiers only will be included to allow unit level analysis. Written consent will not be required for participants of phase 1.

Participants will be provided with contact details for agencies/organisations for self-referral in the event of any emotional distress at any point during participation in this study. Two levels of consent will be obtained for the phase 2 interviews. In the first instance, staff who volunteer to participant in the interviews will be asked to sign a 'consent to contact' slip included in the questionnaire pack. Prior to interview, written informed consent will be obtained. We will use a distress protocol developed for a recent qualitative study involving staff working in critical care during COVID-19 to guide interviewer response if a participant becomes distressed and we will also provide distressed participants access to a self-help psychological well-being toolkit (developed in one NHS Board in Scotland).

### Phase 1 analyses

Demographics of the population will be described using appropriate descriptive statistics (mean (SD), median (IQR) and number (%)). Work-related stress will be compared between the pre-COVID-19 and post-COVID-19 cohorts using the $\chi^2$ test, t-test and analysis of variance (or non-parametric equivalents). Linear and logistic regression analysis will examine the relationship between job demands, job resources, personal resources and demographics on health impairment (burnout syndrome, sleep quality, depression and anxiety), motivation (work engagement and vitality) and organisational outcomes. Structural equation modelling will be used to examine mediating, moderating and latent factors in these relationships.

Two researchers will independently map the support measures identified by staff to the constructs in the JD-R model to identify the theoretical constructs addressed by the support services offered to staff. Kappa will be used to assess agreement. By doing this, the support services that target the sources of stress shown to be significant contributors to personal and occupational outcomes will be identified. Importantly, this process will also identify areas of unmet need.

### Phase 2: theory-based qualitative interviews with CCNs and redeployed nurses

To listen to and understand further CCNs experiences of working in critical care units during the pandemic, we will conduct individual interviews with staff. This will allow more in-depth exploration of the specific issues faced by CCNs in the pandemic including their strategies for dealing with the challenging environment. We plan to recruit around 25 CCNs and around 10 redeployed nurses. We will interview until data saturation is reached using an established stopping criterion.[26]

We will recruit participants from units that reflect the different sizes and patient demographics. If social distancing measures are still in place, we will interview either via an on-line video platform or by phone, according to participant preference. With participants' permission interviews will be digitally voice recorded and transcribed by a University approved transcription service. We will ask participants to describe local supportive initiatives and their views on accessibility, usefulness and effectiveness.

An initial theory-based interview guide will be developed (see online supplemental file), based on the JD-R model and the three theoretical domains known to influence individual and outcomes; job demands, job resources and personal resources. We will use information from our pre-COVID-19 survey, prior 'sense check' interviews with CCNs, and evidence in the available literature to iteratively develop the interview guide; where possible (dependent on timing), we will also use findings from our survey data. Interviews will take approximately 60 min.

## Recruitment

Consenting survey participants will be selected according to a predetermined sampling frame to ensure representation across clinical bandings and other key demographic and professional variables. Informed consent will be obtained and a suitable time for the interview arranged.

## Consent

► A consent to contact form will be included in the questionnaire pack for those who wish to participate in the individual interviews.
► To ensure anonymity of their questionnaire responses, this will be returned in a separate envelope, in the return box and will include the participant's name, banding and contact details.
► A member of the research team will contact the participant to arrange a suitable time for the interview, provide the participant with information and the opportunity to ask questions.
► The interviewer will either read each statement on the consent form to the participant (for telephone interviews) and/or share their screen in online interviews. Verbal consent will be obtained for each statement and will be recorded. Copies of the completed form will be stored securely on university servers.

### *Withdrawal procedure*

Interview participants who wish to withdraw from the study can do so at any time up to the point at which the data are anonymised. Participants who withdraw during the interview will be asked for permission to retain the data already collected anonymously for analysis. Participants who wish to withdraw after the interview is complete can contact the study research fellow (RF) or principal investigator (PI).

## Phase 2 analysis

Transcribed data will be analysed using both deductive and inductive processes. The JD-R model will provide the initial deductive coding framework to which the interview data will be mapped. The remaining interview content will be analysed inductively using the standard framework method[27] to identify and chart content that is not captured by the JD-R model. In this way the adequacy of the JD-R model is examined robustly and its expression in this context will be detailed. Further, the additional content can be evaluated relative to the components of the JD-R model and independently of them. By using this process, the application of theory is preserved, and our qualitative understanding builds in a cumulative theory-based manner. This method will be applied to the data that captures the personal experience of working during the pandemic and content that discusses the experience of the support services offered.

## Integration of data

The application of the JD-R model to both the quantitative and qualitative data facilitates integration of the data. In this way, we will be able to present a unique account of the impact of COVID-19 on this workforce. Comparisons with our pre-COVID-19 data provide an opportunity that will likely not exist elsewhere. In these unparalleled times, we do not yet know whether COVID-19 has had a sustained detrimental effect on this workforce; this is vital information to obtain.

## Data management and data protection

We will comply with the requirements of the General Data Protection Regulations and the Data Protection Act 2018. We will adhere also, where appropriate, to the current version of the NHS Scotland Code of Practice on Protecting Patient Confidentiality. Access to collated participant data will be restricted to the PI and appropriate study staff.

## Study management and oversight

A study management group consisting of the two PIs DD and JR, all coinvestigators and a dedicated RF will meet monthly and co-ordinate the study. The RF will oversee participant recruitment and conduct the individual interviews. In addition, a study steering group will include representation from social science, critical care nursing, and critical care researchers.

## DISCUSSION

There is no doubt that this is a timely and important study. However, we face a number of challenges. At the time of submitting the proposal, indications were that the UK was emerging from the worst of this pandemic. This has been demonstrated not to be the case and one of the challenges will be to recruit participants from an exhausted workforce.

There are a number of limitations. We are recruiting nurses only and therefore are excluding other professional groups and this approach will not give us a full picture of the COVID-19 impact on critical care staff. Our sample also will include nurses from Scotland and England only and are therefore not capturing the experiences of nurses in Wales and Northern Ireland. Although unlikely it may be that their experiences are different.

The main strengths of this study are that we have developed a robust, theoretically informed mixed methods study that will recruit participants from critical care units across both Scotland and England and will represent the range of experiences from those working in small, medium and large units. We will have questionnaire data from a large sample that will be representative of the population and in-depth interview data describing individual experiences of nurses working on the front line during the pandemic. Importantly we will be able to compare work-related stress and its consequences in this professional group with pre-COVID data. This will provide us with unique empirical evidence of the consequences of this pandemic on the critical care and redeployed nurse workforce.

Importantly, this empirical evidence will be actionable. Using a theoretical model of stress will enable the identification of factors that predict stress at a variety of levels. Some of these factors will be amenable to intervention at the level of individual staff and others at higher levels, such as organisational factors. The individual in-depth interviews will augment the quantitative modelling to identify support resources that were helpful, those that were not as helpful and resources that were not offered but might have been helpful. Through a process of matching these resources to the sources of stress we will identify areas of met and unmet need. This actionable understanding of the sources of stress experienced during the pandemic will support more effective targeting of resources to support staff during the course of the current pandemic, should a third wave occur, and in planning for other pandemic scenarios in the future.

**Author affiliations**
[1]Institute of Applied Health Sciences, University of Aberdeen, Aberdeen, UK
[2]Nursing and Health Care, University of Glasgow, Glasgow, UK
[3]Multidisciplinary Adult Psychotherapy Service, NHS Tayside, Dundee, UK
[4]School of Health Sciences, University of Dundee, Dundee, UK
[5]Division of Dietetics, Nutrition and Biological Sciences, Queen Margaret University Edinburgh, Musselburgh, UK
[6]Critical Care, NHS Grampian, Aberdeen, UK
[7]ICU and Anaesthesia, NHS Tayside, Dundee, UK

**Acknowledgements** We would like to acknowledge the support from our current unit contacts: Gill Arbane, Sarah Aucott, Donna Beattie, Clare Brennan, Noreen Clarke, Yvonne Dolan, Jacqueline Gaynor, Andrea Hamilton, Jennifer Howie, Catherine Jardine, Dorothy Kerr, Wendy Laurie, Corrienne McCulloch, Alison McIntosh, Lee McLeish, Louise Murphy, Pauline Murray, Pauline Ringland, Pamela Scott, Deborah Smith, David Watson.

**Contributors** JR, DD, LM, AH, PR, LS, TS and SC contributed to the study protocol. JR, DD, LM, AH, PR, LS, TS, SC and JM contributed to the manuscript.

**Funding** The study is funded by the National Institute for Healthcare Research 132068.

**Competing interests** None declared.

**Patient consent for publication** Not required.

**Provenance and peer review** Not commissioned; externally peer reviewed.

**ORCID iD**
Janice Rattray http://orcid.org/0000-0002-0563-8498

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
