## [Reviewer comments · BMJ Open]

ARTICLE DETAILS

TITLE (PROVISIONAL)	Work-related stress: The Impact of COVID-19 on Critical Care and Redeployed Nurses: A mixed methods study
AUTHORS	Rattray, Janice; McCallum, Louise; Hull, Alastair; Ramsay, Pam; Salisbury, Lisa; Scott, Teresa; Cole, Stephen; Miller, Jordan; Dixon, Diane

VERSION 1 – REVIEW

REVIEWER	McKnight, Jacob University of Oxford, Nuffield Department of Medicine
REVIEW RETURNED	02-Apr-2021

GENERAL COMMENTS	I think the research described is very much needed and the approach described is highly robust. One small note: if during interviews nurses become very distressed or upset, it may be good to have more than a referral system in place. Where real issues are encountered as part of the research, I think the interviewer/s probably has a duty of care to ensure the respondent actually gets care. The process for dealing with highly distressed individuals wasn't entirely clear in the proposal.
---

REVIEWER	Vallone, Federica University of Naples Federico II, Humanities
REVIEW RETURNED	02-Apr-2021

GENERAL COMMENTS	This protocol entails a two-phase mixed methods study (QUANT – QUAL), underpinned by the well-known JD-R Model, which aims at understanding the impact of COVID-19 on Critical Care (CCNs) and redeployed nurses and NHS organisations. Despite the study sought to target a pivotal issue by aiming at exploring the impact of COVID-19 on nursing staff wellbeing, unfortunately, I have several concerns about its rationale. Indeed, the authors stated that “Comparisons with our pre-COVID-19 data provide an opportunity that will likely not exist elsewhere. In these unparalleled times, we do not yet know whether COVID-19 has had a sustained detrimental effect on this workforce; this is vital information to obtain”. However, it did not explain the reason why it could be useful to replicating a pre-COVID-19 study. In particular, the authors stated that nurses “have been working in a highly charged environment with additional challenges”. Nevertheless, by comparing results with a pre-covid situation, these new and unique challenges could hardly be addressed and the sole integration of data from qualitative analyses could partially provide customized information about demands and resources (from JD-R) for developing evidence-based
--

	interventions promoting health among nursing professionals. Additionally, the protocol is not clear:  1) the protocol lack in explaining in detail the pre-COVID-19 study which was already carried out (the JD-R Model has not a fixed framework in terms of operationalized variables). 2) Hypotheses and practical implications this study could have in fostering individual and organizational wellbeing are missing. 3) There is no rigour in presenting the study variables and measurement tools. Given as examples: in the abstract as well as in the first part of the paper, it seems that health outcomes will be 1) burnout syndrome 2) post-traumatic stress symptoms. However, Outcome Measure section also includes the assessment of "Mental health using the GHQ-12 (12 items)". Moreover, Motivation section also includes "Other individual outcomes including 'Turnover' (3-items), 'Recovery after Work' (6-items), 'Detachment after Work' (3-items), that will be assessed by means of the QEEW 2.0". I suggest the authors to re-consider this way to organize the variables included within the framework. For examples, I wonder why "turnover" is not addressed among the organizational outcomes and why "Recovery after Work" and "Detachment after Work" are theoretically included among the motivational factors as well as among the outcomes only? Finally, explanations about the statistical analyses that will be carried out should be integrated and the reference list currently lacks in addressing research conducted over the past year on the topic. In conclusion, I suggest the author(s) carry out a major revision of the manuscript, in order to strengthen its clear potential.
--	--

VERSION 1 – AUTHOR RESPONSE

Reviewer comment	Response
One small note: if during interviews nurses become very distressed or upset, it may be good to have more than a referral system in place. Where real issues are encountered as part of the research, I think the interviewer/s probably has a duty of care to ensure the respondent actually gets care. The process for dealing with highly distressed individuals wasn't entirely clear in the proposal.	Thank you for making this point. We are using a distress protocol, developed recently for a qualitative study exploring staff experiences during COVID. This involves clear guidance for the interviewer in how to support distressed staff. Actions include when to stop the interview, remain virtually with the participant until their distress abates or decreases, offer to contact family or friends, and gain permission to contact the next day to ensure they are no longer distressed. We have referred to this protocol albeit not with this detail in the text. I hope you are reassured we are very aware of this potential.
Comparisons with our pre-COVID-19 data provide an opportunity that will likely not exist elsewhere. In these unparalleled times, we do not yet know whether COVID-19 has had a sustained detrimental effect on this workforce; this is vital information to obtain". However, it did not explain the reason why it could be useful to replicating a pre-COVID-19 study. In particular, the authors stated that nurses "have been working in a highly charged environment with additional	Thank you for making these helpful comments. We have strengthened the rationale for using pre-COVID data within the text. We refer to this in the text by explaining that our comparator pre-COVID study used most of the proposed measures, and that by having this comparator will give us improved understanding of the impact of the pandemic. The qualitative element will also increase our understanding of the lived experiences of critical care nurses. This is not a complete replication study but one where we will capture staff experiences at a time

challenges". Nevertheless, by comparing results with a pre-covid situation, these new and unique challenges could hardly be addressed and the sole integration of data from qualitative analyses could partially provide customized information about demands and resources (from JD-R) for developing evidence-based interventions promoting health among nursing professionals.	when they are still able to recall these experiences with some clarity and we are interested mainly in the potential enduring consequences for the individual and NHS.
The protocol lack in explaining in detail the pre-COVID-19 study which was already carried out (the JD-R Model has not a fixed framework in terms of operationalized variables).	We have included a sentence about our pre-COVID-19 study (a PhD study nearing completion) in the introduction but have chosen not to give the detail as much of this is the same as our current design. Results from this pre-COVID study will be presented with the results from this study.
Hypotheses and practical implications this study could have in fostering individual and organizational wellbeing are missing.	We have chosen not to include hypotheses but prefer to remain with our stated research questions. We have added suggested practical implications of the study to the end of the Discussion section of the protocol.
There is no rigour in presenting the study variables and measurement tools.	We have included additional detail about each of the proposed measures.
In the abstract as well as in the first part of the paper, it seems that health outcomes will be 1) burnout syndrome 2) post-traumatic stress symptoms. However, Outcome Measure section also includes the assessment of "Mental health using the GHQ-12 (12 items)". Moreover, Motivation section also includes "Other individual outcomes including 'Turnover' (3-items), 'Recovery after Work' (6-items), 'Detachment after Work' (3-items), that will be assessed by means of the QEEW 2.0". I suggest the authors to re-consider this way to organize the variables included within the framework. For examples, I wonder why "turnover" is not addressed among the organizational outcomes and why "Recovery after Work" and "Detachment after Work" are theoretically included among the motivational factors as well as among the outcomes only? Finally, explanations about the statistical analyses that will be carried out should be integrated and the reference list currently lacks in addressing research conducted over the past year on the topic. In conclusion, I suggest the author(s) carry out a major revision of the manuscript, in order to strengthen its clear potential.	We have amended the introduction and body of the text to: include mental health, recovery from work and detachment from work within the health outcomes, and placed turnover and patient safety culture in the organisational outcomes. We appreciate your comment about contemporary literature, and we have included Greenberg's paper. We are monitoring the current literature but feel uncomfortable about adding any additional references as we submitted this protocol and proposal before these were available and we are reluctant to add these and thereby suggest this literature informed the proposal and protocol.

VERSION 2 – REVIEW

REVIEWER	Vallone, Federica University of Naples Federico II, Humanities
REVIEW RETURNED	23-May-2021
GENERAL COMMENTS	Thank you for the possibility to revise once again this interesting protocol. I appreciate that the authors addressed most of my suggestions. I have only one further suggestion about the Title. Please consider re-wording the title. Indeed, it contains colons (:) two times. Good luck with your work.